# Efficient biosynthesis of heterodimeric C$^3$-aryl pyrroloindoline alkaloids

Wenya Tian[1], Chenghai Sun[1], Mei Zheng[1], Jeffrey R. Harmer[2], Mingjia Yu[3], Yanan Zhang[1], Haidong Peng[1], Dongqing Zhu[1], Zixin Deng[1], Shi-Lu Chen[3], Mehdi Mobli [2], Xinying Jia [2] & Xudong Qu [1]

Many natural products contain the hexahydropyrrolo[2, 3-b]indole (HPI) framework. HPI containing chemicals exhibit various biological activities and distinguishable structural arrangement. This structural complexity renders chemical synthesis very challenging. Here, through investigating the biosynthesis of a naturally occurring C$^3$-aryl HPI, naseseazine C (NAS-C), we identify a P450 enzyme (NascB) and reveal that NascB catalyzes a radical cascade reaction to form intramolecular and intermolecular carbon–carbon bonds with both regio- and stereo-specificity. Surprisingly, the limited freedom is allowed in specificity to generate four types of C$^3$-aryl HPI scaffolds, and two of them were not previously observed. By incorporating NascB into an engineered strain of *E. coli*, we develop a whole-cell biocatalysis system for efficient production of NAS-C and 30 NAS analogs. Interestingly, we find that some of these analogs exhibit potent neuroprotective properties. Thus, our biocatalytic methodology offers an efficient and simple route to generate difficult HPI framework containing chemicals.

[1] Key Laboratory of Combinatorial Biosynthesis and Drug Discovery, Ministry of Education, School of Pharmaceutical Sciences, Wuhan University, 430071 Wuhan, China. [2] Centre for Advanced Imaging, The University of Queensland, St. Lucia, QLD 4072, Australia. [3] School of Chemistry and Chemical Engineering, Beijing Institute of Technology, 100081 Beijing, China. These authors contributed equally: Wenya Tian, Chenghai Sun. Correspondence and requests for materials should be addressed to X.J. (email: x.jia1@uq.edu.au) or to X.Q. (email: quxd@whu.edu.cn)

common heterocyclic motif observed in a large number of alkaloids and synthetic compounds is hexahydropyrrolo[2, 3-*b*]indole (HPI), usually referred to as pyrroloindoline. Pyrroloindoline-containing natural products exhibit a broad array of biological properties, ranging from anticancer and antibacterial activities to the inhibition of cholinesterase[1]. Naturally occurring $C^3$-aryl pyrroloindolines are mostly manifest in the fungal-sourced, tryptophan-based homodimeric diketopiperazine (DKP), in which two pyrroloindoline units are fused in a $C^3$–$C^{3'}$ bond (Fig. 1)[1–3]. Unlike the homodimeric DKPs that are in large abundance (>100 members)[1–3], only five heterodimeric DKPs have been identified so far, three of which are naseseazine A, B, and C (NAS-A, B, and C) produced by a bacterial system (Fig. 1)[4–6]. Except that L-Ala in NAS-A is replaced by L-Pro in NAS-B, NAS-A is identical to NAS-B. In NAS-B and C, the identical pyrroloindoline and DKP moieties are connected in two different ways: (i) the $C^3$-aryl pyrroloindoline framework is formed through a $C^3$–$C^{7'}$ bond and with 2*R*-3*S* stereo-configuration (NAS-B); (ii) the connection is formed through a $C^3$–$C^{6'}$ bond and with 2*S*-3*R* chirality (NAS-C).

The characteristic molecular architecture and promising medicinal value of these products have garnered extensive interest particularly with respect to efforts to develop a variety of chemical methods for enantio-selective synthesis of pyrroloindoline-containing products[7]. However, the regio- and stereo-specificity in the densely functionalized frameworks of NASs, especially at the quaternary stereocenter at the $C^3$ position that includes an aryl substituent, requires tremendous efforts in its chemical preparation through organic synthesis[8,9]. This feature severely impedes an assessment of structural diversity and associated biological activities of NASs. In the chemical synthesis of NASs, regio-specificity was achieved by pre-installation of direction groups in both of the pyrroloindoline and DKP moieties, resulting in long synthetic steps and very low yields[7,9]. The only stereo-configuration accomplished is the $C^2R$-$C^3S$, which is induced by the $C^{11}$ stereocenter (derived from C$\alpha$ of tryptophan) (Fig. 1)[7]. However, this induction from the $C^{11}$ stereocenter is not able to generate the unfavored $C^2S$-$C^3R$ in NAS-C. So far, no

successful strategies have been reported to chemically synthesize NAS-C. As a practical alternative to chemical synthesis of NASs, two *Streptomyces* strains have been reported to produce NAS-A, -B, and -C[4–6]. The biosynthesis of NAS-A, -B, and -C thus may provide an enzyme-catalyzed route for the generation of NASs through a stereo- and regio-chemically defined reaction.

Here, we unveil the biosynthesis of NAS-C and discover a key P450 enzyme, which catalyzes a highly regio- and stereo-selective $C^3$-aryl bond-forming step to generate NAS-C. We further incorporate this P450 enzyme into a highly efficient whole-cell biocatalysis system. This engineered whole-cell factory is fed with synthetic cyclodipeptides to produce 30 heterodimeric $C^3$-aryl pyrroloindolines (NASs analogs). Finally, some of those NASs analogs are found to have potent neuroprotective properties.

## Results

**Deciphering the biosynthesis of NASs.** Fijian marine-sourced *Streptomyces* sp. CMB-MQ030 (MQ030 strain) was previously reported to produce only NAS-A/B[4,6], so our initial aim was to identify the biosynthetic gene clusters of NAS-A/B. We hypothesized that NAS-A/B were assembled by two molecules of cyclodipeptides (2,5-diketopiperazine) through a radical mechanism. In order to locate the biosynthetic gene clusters, we sequenced and assembled the draft genome of *Streptomyces* sp. CMB-MQ030 (8,454,906 bp). Analysis of the draft genome of the MQ030 strain revealed that, neighbored by several genes for regulation and exportation, each of three distinct loci (locus-1, 2, and 3) contains one tRNA-dependent cyclodipeptide synthase (CDPS) gene and one adjacent *P450* gene, which are functionally competent for a hypothesized biosynthetic route (Fig. 2a). Locus-1 and locus-2 share high similarity to each other: 61% identity between two CDPS genes and 68% identity between the two *P450* genes; locus-3 harbors additional albonoursin biosynthetic genes[10] (CDO-A and B) and is therefore excluded from further consideration.

To validate that locus-1 or -2 is the biosynthetic gene cluster of NAS-A/B, heterologous expression of locus-1 and -2 was performed in *Streptomyces albus* J1074: no obvious metabolite was detected in the recombinant strain with locus-2, whereas the recombinant strain with locus-1 resulted in production of a single metabolite with identical molecular weight to NAS-B (Fig. 2b, trace IV). However, the high performance liquid chromatography (HPLC) retention time of this metabolite is different from that of standard NAS-B in HPLC, indicating that this metabolite is not NAS-B (Fig. 2b, trace V). This metabolite was then determined to be NAS-C by comparing the nuclear magnetic resonance (NMR) data with the reported ones[5]. Surprisingly, the production of NAS-C in MQ030 strain was not detected (Fig. 2b, trace III), though both NAS-C and NAS-A/B are produced by Australian marine-sourced *Streptomyces* sp. USC-636[5]. As locus-1 only produce NAS-C, we infer that NAS-A/B must be encoded by a different biosynthetic pathway, i.e., locus-2, though heterologous expression of locus-2 in host *S. albus* failed to produce NAS-A/B.

With decoupling from CDPS, the in vitro activity of P450 enzymes in locus-1 and -2 was also assayed to verify that the P450 works on the products of CDPS to produce NASs. It is well-characterized that CDPS catalyzes the formation of cyclic peptide dimers (cyclodipeptides), enabling us to propose that those cyclodipeptides would be substrates of the P450. Therefore, we directly feed synthetic cyclodipeptides into the P450-catalyzed reaction to assay the activity of the P450. Neither *P450* genes from locus-1 nor -2 could be expressed in *Escherichia coli* BL21 (DE3), until their gene codons were optimized for *E. coli* usage. P450-1 (P450-NAS-C or NascB) is partially soluble in the supernatant and can be successfully purified, while P450-2 forms

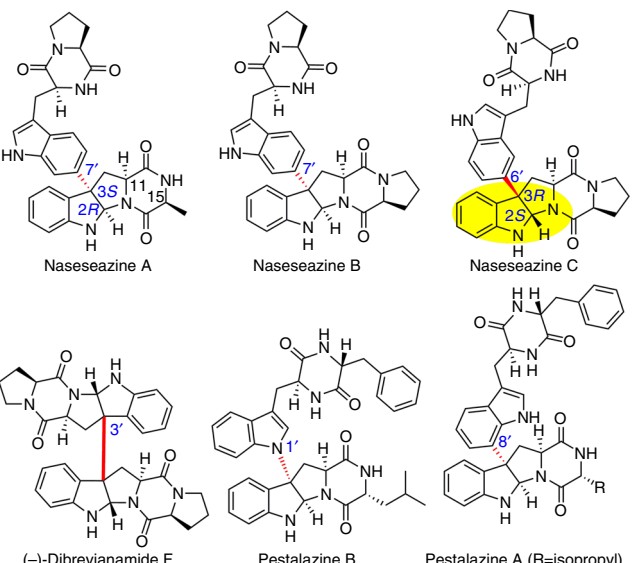

**Fig. 1** The structures of some naturally occurring $C^3$-aryl pyrroloindolines. Highlighted in nasseazine C is pyrroloindoline motif featuring 2*S*-3*R* chirality and $C^3$–$C^{6'}$ bond connectivity. Other types of $C^3$-aryl connectivity are highlighted with a red bond

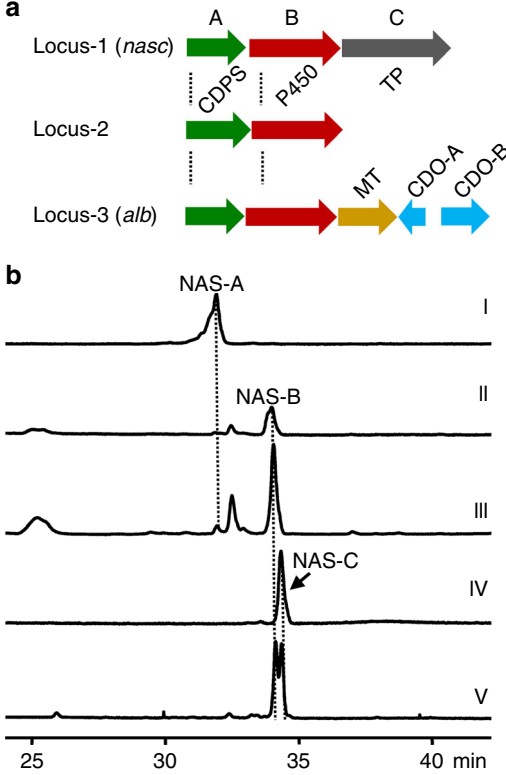

**Fig. 2** Gene clusters and correlation of locus-1 to NAS-C biosynthesis. **a** Biosynthetic gene clusters with CDPS and P450 in CMB-MQ030. Locus-1 and 2 are resembling only except locus-1 contains an additional TP (transporter) gene; locus-3 containing MT (methyltransferase), CDO-A (cyclodipeptide oxidase A), and CDO-B (cyclodipeptide oxidase B) is homologous to *alb* biosynthetic gene cluster. **b** HPLC analysis the production of NAS-C in CMB-MQ030 and locus-1 expression *S. albus*. (I) Standard NAS-A, (II) standard NAS-B, (III) CMB-MQ030, (IV) mWHU2475 (*S. albus* containing locus-1), and (V) standard NAS-C spiked with standard NAS-B from the co-injection of standard NAS-C and -B

inclusion bodies and was thus not characterized further. In the presence of *E. coli*-sourced flavodoxin (Fdx), flavodoxin reductase (FdxR) and an NADPH recycle system (NADP, glucose, and glucose dehydrogenase), P450-NAS-C (NascB) can convert the synthetic cyclo-(L-tryptophan-L-proline) (**s4**, $cW_L$-$P_L$) into NAS-C (Fig. 3a, trace III). Consistent with the in vivo result that no NAS-B was produced in this reaction, this enzyme is unequivocally confirmed to only be responsible for NAS-C biosynthesis.

**Delineating the mechanism of the P450 reaction.** Typical P450 reactions require the substrates to enter the active site of the P450, but not directly bind to the $Fe^{III}$ ion[11,12]. To investigate if substrate-binding causes a change in the electronic environment of the enzyme ferric heme and hence binds to the NascB, X-band continuous wave electron paramagnetic resonance (CW EPR) was used to measure the ferric heme signal. The spectra recorded at 15 K for the substrate-free enzyme (Fig. 3b, top traces) was very similar to that previously reported[11], showing the enzyme predominantly in the low-spin state (LS) due to axial water coordination. The spectrum comprises a number of EPR components as indicated by the simulation (Fig. 3b, red trace) due to small differences in the orientation of the coordinated water molecule. Upon addition of $cW_L$-$P_L$, the majority of the ferric heme iron remains LS (Supplementary Figure 1), but only a single species is observed with shifted *g*-values, confirming a modification of the

heme iron electronic environment due to the substrate $cW_L$-$P_L$ binding to the protein.

While P450 can either catalyze oxygenation or radical-mediated coupling reactions, we hypothesize that the mechanism of forming NAS-C is a radical-mediated coupling reaction. This is supported by our result that the reaction process can be gradually inhibited by the increasing concentration of the radical scavenger TEMPO (Supplementary Figure 2). Similar to NAS-C, the biosynthesis of (-)-dibrevianamide F (homodimeric DKP) from the fungus *Aspergillus flavus* (*A. flavus*) was also assumed through a radical mechanism[13]. Instead of CDPS, *A. flavus* uses nonribosomal peptide synthase (NRPS) to synthesize the cyclodipeptide substrate and a P450 enzyme, DtpC, for catalyzing the dimerization. DtpC is proposed to initiate the reaction by abstracting a hydrogen from $N^{10}$ or $N^{1'}$ of the $cW_L$-$P_L$ substrate to form $N^{10}\bullet$ radical[13] or $N^{1'}\bullet$ radical[14]. $N^{10}$-radical ($N^{10}\bullet$) undergoes intramolecular addition to $C^2$ to form the pyrroloindoline $C^3$-radical. Two pyrroloindoline $C^3$-radicals react with each other to yield (-)-dibrevianamide F, the major homodimeric product (Supplementary Figure 3)[13]. For two marginal hetero-dimeric heterodimeric DKPs, $N^{1'}$-radical ($N^{1'}\bullet$) can either directly couple with the pyrroloindoline $C^3$-radical to form the $C^3$–$N^{1'}$ bond, or migrates to $C^{7'}$ first and then couple with the pyrroloindoline $C^3$-radical to generate NAS-A/B (Supplementary Figure 3)[14].

Using the density functional theory (DFT) calculations (Supplementary Methods), the structures of $N^{10}\bullet$ and $N^1\bullet$ radical have been optimized (Supplementary Figure 4). It is revealed that the unpaired spin density in $N^1\bullet$ is more delocalized than in $N^{10}\bullet$ (Supplementary Figure 4, Supplementary Dataset). Consequently, $N^1\bullet$ is more stabilized and has a free energy ($\triangle G$) 16 kcal mol$^{-1}$ lower than $N^{10}\bullet$, implying that P450 enzymes most likely prefer the hydrogen abstraction from the $cW_L$-$P_L$ $N^1$ atom (instead of $N^{10}$) to form the $N^1\bullet$. H-atom abstraction, rather than single electron transfer, is also supported by thiolate ligation in P450 through significantly decreasing heme reduction potential and elevating pKa of compound II[15,16]. To further support this mechanism, we prepared an oxo-mimic of $cW_L$-$P_L$ (Oxo-$cW_L$-$P_L$) (**s30** in Supplementary Figure 5), in which the $HN^1$ was replaced by an oxygen. Biochemical assay revealed that this substrate is not able to be converted by the NascB, suggesting $HN^1$ is critical for the enzymatic reaction. As assumed in the biosynthesis of communesin[17], calycanthine, and chimonanthine[18,19], $N^1\bullet$ could migrate to $C^3$, followed by a Mannich-type reaction occurring between the $N^{10}$ and imine bond of $N^1$-$C^2$. In addition, a similar mechanism has been recently proposed in chemical oxidation of tryptophan[20]. Collectively, we can conclude that the $N^1\bullet$-mediated intramolecular Mannich reaction most probably results in the formation of pyrroloindoline $C^3$ radical (Fig. 3c). Afterwards, $C^3$ radical could undergo two possible routes: (1) the radical inserts into $C^{6'}$ of another $cW_L$-$P_L$ followed by elimination to generate NAS-C; (2) the $C^3$ radical turns into $C^3$ cation, which attaches $C^{6'}$ of another $cW_L$-$P_L$ to form NAS-C under a Friedel–Crafts scheme. We rule out the second route with Friedel–Crafts scheme of electrophilic aromatic substitution, because NascB-catalyzed reaction rates are not affected by strong electron-withdrawing group F on 7-F substituted $cW_L$-$P_L$ (Fig. 3c, **NAS-27** in Fig. 4, and Supplementary Figure 6).

NAS-C bears a distinct bond of $C^3$–$C^{6'}$. Given that the proposed $C^{6'}$ radical can't be generated from the migration from $N^{1'}$, the $C^3$–$C^{6'}$ bond is more likely to be formed through an intermolecular (for $C^{6'}$) radical additions, in which the nascent pyrroloindoline $C^3$ radical directly attacks the $C^{6'}$ of the second molecule of cyclodipeptide to form the bond (Fig. 3c). To confirm

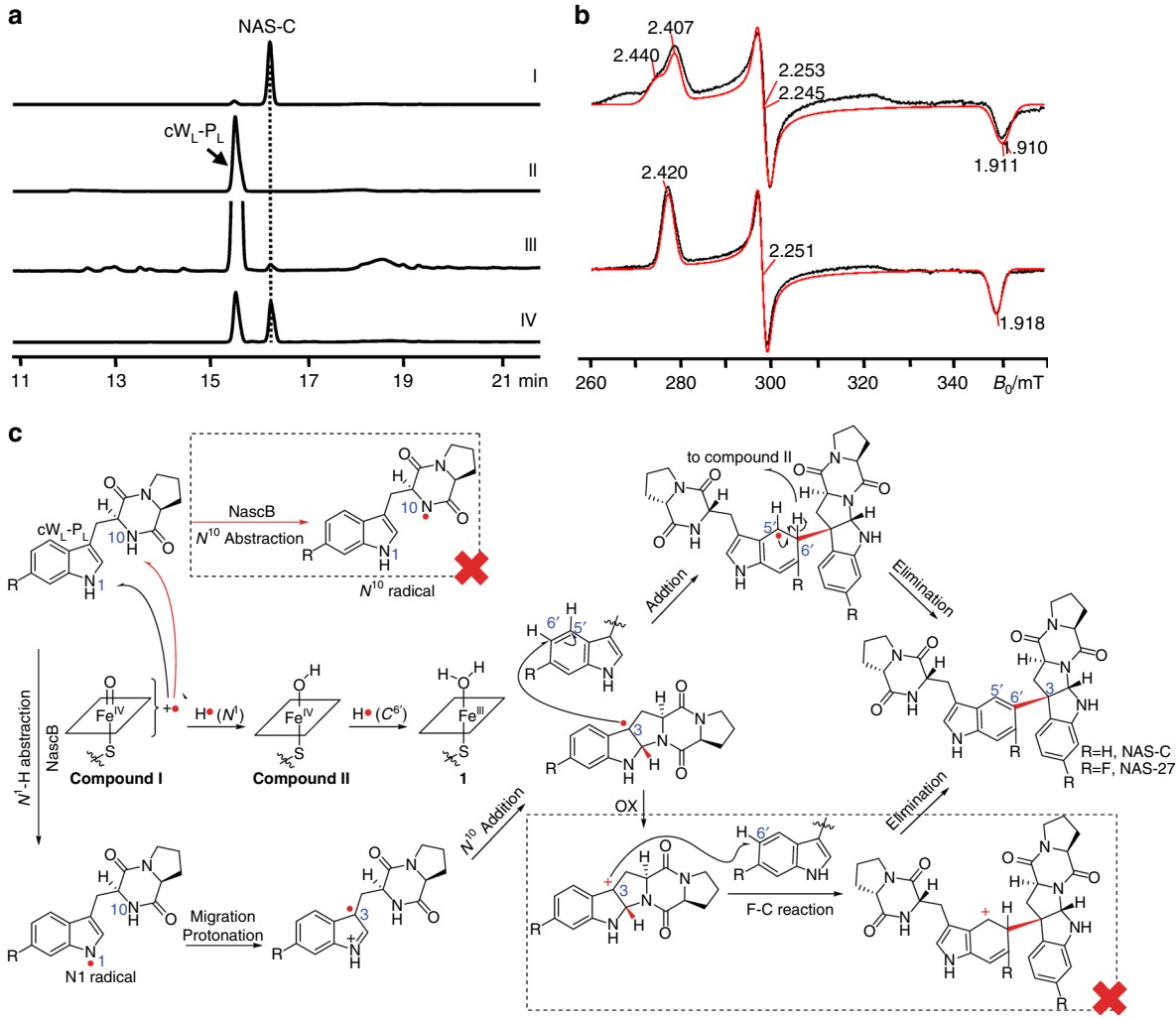

**Fig. 3** Characterization of NascB in vitro and the proposed mechanism. **a** HPLC analysis of the NascB-catalyzed reaction. (I) Standard NAS-C, (II) a control reaction, the same as NascB reaction but only omitting the P450 NascB, (III) NascB reaction, using *E. coli* Fdx and FdxR as electron supply system, (IV) NascB reaction, using spinach Fd and FdR as electron supply system. **b** X-band (9.3810 GHz) CW EPR spectra recorded at 15 K showing the low-spin (LS) ferric heme signal in the absence (the upper panel) and presence (the lower panel) of the substrate (twofold excess of cW$_L$-P$_L$). Simulations computed using the principal *g*-values are shown in red: (the upper panel) two component model with *g* = (2.407, 2.253, 1.911) and *g* = (2.440, 2.245, 1.910); (the lower panel) single component model, *g* = (2.420, 2.251, 1.918). **c** Proposed mechanisms of the radical-mediated intramolecular cyclization and intermolecular addition reaction by NascB. The NascB mechanism is proposed to involve the active species compound I (Fe$^{IV}$ = O+•)[24-26,34]. It abstracts a proton with an electron from the HN$^1$ of the substrate to form the active species compound II (Fe$^{IV}$(-OH)), which is further converted into the ferric species (**1**) by accepting a hydrogen radical from C6′ position of the substrate to form one water molecule. The water coordination turns NascB into the resting state P450, completing the catalytic cycle[12,35]. A significant high Gibbs free energy (△*G*) calculated by DFT calculation rule out the possibility of N$^{10}$ radical (marked by a dashed rectangle with a red cross). The possibility of Friedel–Crafts reaction from pyrroloindoline C$^3$-radical was ruled out by the insensitivity of NascB-catalyzed reaction to the electron-withdrawing group introduced at C$^7$-position of the substrates

this mechanism, we synthesized and assayed a variety of cyclodipeptide substrates to seek the co-generation of products with different regio-selectivity. NascB transformed the substrate cyclo-(L-tryptophan-L-valine) (cW$_L$-V$_L$) (**s3**, Supplementary Figure 5) into both products with C$^3$–C6′ (**NAS-18**) and C$^3$–C7′ (**NAS-17**) bond, respectively. Similar outcomes were also observed in **NAS-4/3**, **6/5**, **20/19**, **22/21** and **24/23** (Fig. 4). The variation of observed bonds cannot be generated by radical coupling, but rather from the addition of a pyrroloindoline C$^3$ radical to either the C6′- or C7′ position. Based on these results, we confirmed that the formation of NAS-C involves a mechanism of radical-mediated intramolecular cyclization and intermolecular addition (Fig. 3c), which is very efficient for the synthesis of complex natural products.

**Developing an *E. coli*-based whole-cell biocatalysis system**. The in vitro catalyzed system requires the tedious purification of multiple enzymes, supplementation with the expensive cofactor NADPH and it is not suitable for large-scale preparation, so we decided to develop an *E. coli*-based whole-cell biocatalysis system for NAS synthesis. The NAS-C reaction depends on electron transportation systems and, therefore, we first evaluated different pairs of electron transport systems to optimize this reaction. As NAS-C can be produced by heterologous expression in *S. albus*, the endogenous ferrodoxin (Fd) and ferrodoxin reductase (FdR) of *S. albus* could be competent for this reaction. Thereby, all four FdR and three Fd genes from *S. albus* were amplified and cloned into pET28a for expression in *E. coli*. Unfortunately, none of Fd and FdR combination show activity and thus were excluded for

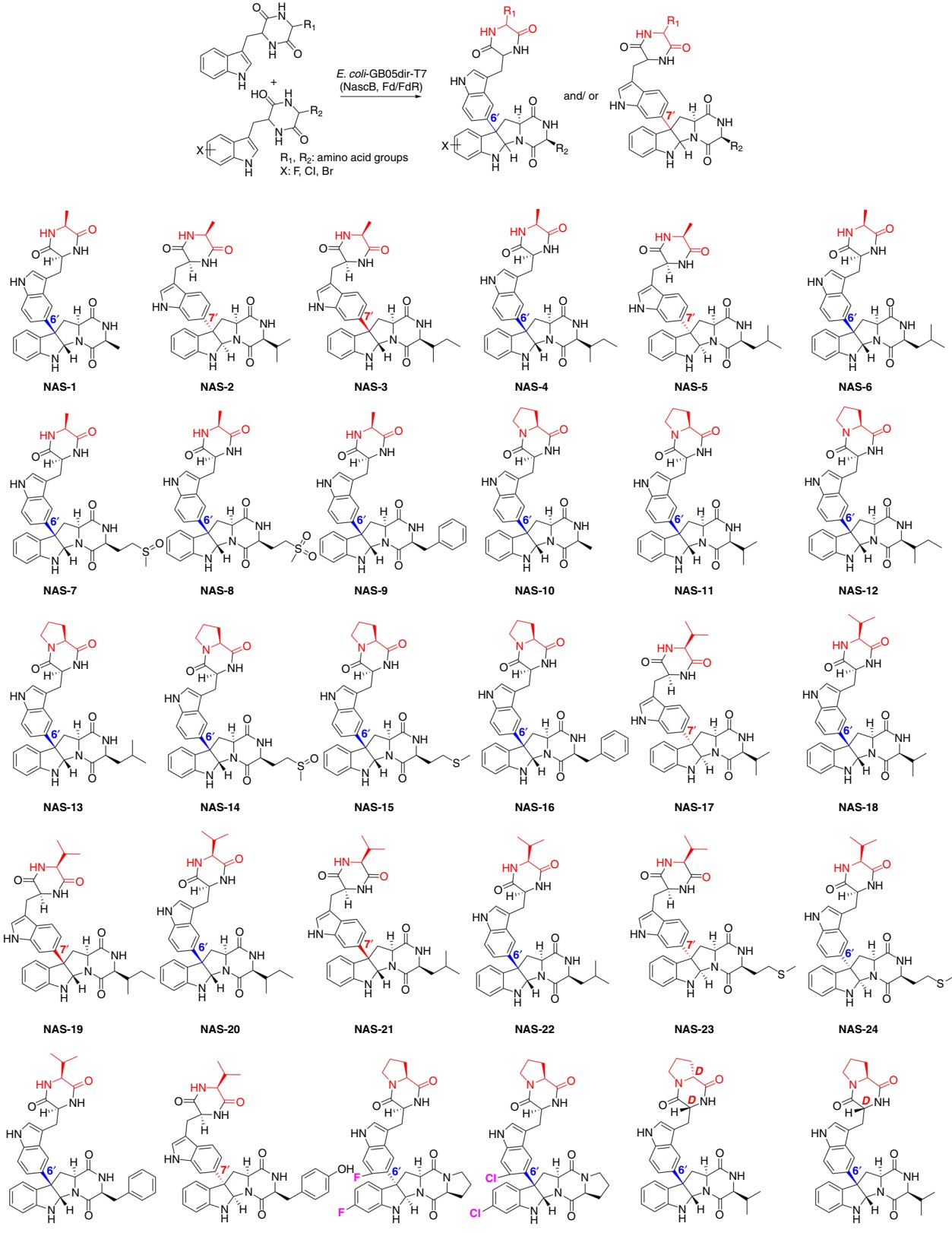

**Fig. 4** Products generated by the catalysis of the whole-cell system. The non-tryptophan residues in the upper moiety are highlighted in red. Connections between the upper moiety and lower pyrroloindoline moiety are indicated by two different colors: blue color for C³–C⁶′ bond, while red color for C³–C⁷′ bond

further study. Although we could use the *E. coli*-sourced Fdx and FdxR, the commercial *Spinach* Fd and FdR were tested and found to provide much better activity than the *E. coli*-sourced Fdx and FdxR (~50-fold based on the conversion yield, Fig. 3a, trace IV).

Both spinach *fd* and *fdr* genes were synthesized with codons optimized for *E. coli* and constructed into a variety of plasmids for evaluation of their expression yield in *E. coli*. When fused with a TRX and MBP tag at N-terminal, respectively, Fd and FdR protein can be expressed well, and the fusion proteins are fully competent without a need to remove the tags. Finally, we cloned *trx-fd* and *mbp-fdr* into a pRSFduet vector and *nascB* into pET21a to achieve the co-expression of Fd, FdR, and NascB in *E. coli*.

Unexpectedly, co-expression of these genes (*fd*, *fdr*, and *nascB*) in *E. coli* BL21 (DE3) showed a highly toxic effect as cells rapidly underwent self-lysis after the expression was induced by IPTG. As cells are safe upon the individual expression of Fd, FdR, and NascB, the toxicity effect must be derived from the activated NascB in the presence of both Fd and FdR. After screening several commercial *E. coli* strains including *E. coli* Rosetta (DE3), BL21 (DE3)-pLysE, C41 (DE3), and C43 (DE3), which are widely used and some claimed to be particularly suitable for toxic protein expression, none of those strains can survive the activated NascB.

All above tested *E. coli* strains are a derivative of *E. coli* BL21 (DE3), a B type strain. The B type strains are often used for protein expression, while K type strains are mostly used for DNA cloning but also for protein expression, such as Shuffle T7 competent *E. coli* from New England Biolabs. Considering the difference between these two types, K type strain may be tolerant to the toxicity of activated NascB. We further screened two different K strains, i.e., the commercial strain *E. coli* JM109 (DE3) and a homemade strain *E. coli* GB05dir-T7. *E. coli* GB05dir is a derivative of DH10B by integration of RecET proteins on the genome[21]. For satisfying the requirement of expressing Fd, FdR, and NascB under T7 promoter, a T7 polymerase-coding gene was integrated into the *lac* operon, yielding *E. coli* GB05dir-T7. Surprisingly, the resulting strain *E. coli* GB05dir-T7 is very robust for the complete P450 catalytic system expression, while *E. coli* JM109 (DE3)-T7 died rapidly as other B type strains did. Both *E. coli* JM109 (DE3) and *E. coli* GB05dir are derived from the prototype strain K12, so their genotype is similar. The most obvious difference is the deficiency of the Lon protease in JM109 (DE3). Like other B type (DE3) strains, the *lon* gene was knocked out as it causes protein degradation. However, Lon protease is also required for stress-induced developmental changes and survival from DNA damage[22,23]. Because activated P450 can generate radical species, which may be able to cause DNA damage, we assume the *lon* deficiency in *E. coli* strains could be the major reason for cell death. Considering Lon can cause protein degradation, the expression yield for every single protein in GB05dir-T7 and BL21 (DE3) was compared to show the effect of Lon on protein expression is trivial: the protein yield in GB05dir-T7 is only slightly lower (~20%) than BL21 (DE3). Therefore, this strain is still efficient for protein expression and could be suitable for many other toxic P450 systems. Using this *E. coli* GB05dir-T7-based whole-cell system, complete conversion of cW$_L$-P$_L$ into NAS-C can be achieved by an overnight incubation (Supplementary Figure 6a). Considering that the P450 reaction requires NADPH, we further tried co-expression of glucose dehydrogenase (GDH) with Fd, FdR, and NascB in GB05dir-T7, but the catalytic activity of NascB did not improve, suggesting that the endogenous NADPH supply in the *E. coli* is indeed sufficient.

**Generation of structural varieties through biocatalysis.** After establishing the cell biocatalysis system, we set out to perform biocatalysis to generate NAS varieties by feeding *E. coli* cells with 20 chemically synthesized and L-Trp containing cyclodipeptide (Supplementary Figure 5), i.e., cW$_L$-X$_L$, where X$_L$ denotes one of 20 natural L-amino acids. To evaluate the substrate specificity of NascB, these cyclodipeptides except for the natural cW$_L$-P$_L$ were individually fed to the recombinant *E. coli* (GB05dir-T7) containing the *nascB*, *trx-fd*, and *mbp-fdr* genes (GB05dir-T7-NascB). Three products were generated in high yield (Fig. 4, Supplementary Table 1, and Supplementary Figure 6b, c), including a cW$_L$-A$_L$ dimerization (**NAS-1**) and two cW$_L$-V$_L$ dimerization products (**NAS-17** and **NAS-18**). Interestingly, **NAS-1** and **NAS-18** have identical connections and stereo-configuration as in NAS-C (C$^3$–C$^{6'}$ and 2S-3R), while **NAS-17** resembles NAS-A/B (C$^3$–C$^{7'}$ and 2R-3S). The production of **NAS-17** suggested that NascB indeed also has a relaxed regio- and stereo-specificity in addition to its broad spectrum of substrates.

The efficient generation of NAS-C, **NAS-1**, **17**, and **18** suggests that substrates cW$_L$-P$_L$, cW$_L$-A$_L$, and cW$_L$-V$_L$ can be accepted readily by NascB. Considering that each of the pyrroloindolines is formed by two units of cyclodipeptides, we were interested in combining these three substrates with other cyclodipeptides to generate hetero-pyrroloindolines. Following this aim, each of these three substrates was individually co-fed with one of the remaining 17 cyclodipeptides into the recombinant *E. coli* GB05dir-T7-NascB. To our gratification, besides the production of **NAS-C**, **-1**, **-17**, and **-18** (homo-dimerization), this co-feeding of two different cyclodipeptides at one time resulted in additional 23 products (hetero-dimerization, Fig. 4, Supplementary Tables 2–4 and Supplementary Figures 7–9): 8 cW$_L$-A$_L$-derived hetero-pyrroloindolines (**NAS-2** to **NAS-9**), 7 cW$_L$-P$_L$-derived hetero-pyrroloindolines (**NAS-10** to **NAZ-16**), and 8 cW$_L$-V$_L$-derived hetero-pyrroloindolines (**NAS-19** to **NAS-26**). Interestingly, the selectivity of the upper cyclodipeptide is much stricter than the lower pyrroloindoline moiety; only cW$_L$-P$_L$, cW$_L$-A$_L$, and cW$_L$-V$_L$ can be accepted as the upper moiety. Furthermore, when co-feeding cW$_L$-P$_L$ and cW$_L$-A$_L$ or cW$_L$-V$_L$, cW$_L$-P$_L$ was accepted as the upper moiety (**NAS-10** and **11**); when co-feeding cW$_L$-A$_L$ and cW$_L$-V$_L$, cW$_L$-A$_L$ was accepted as the upper moiety (**NAS-2**), suggesting that cW$_L$-P$_L$, cW$_L$-A$_L$, and cW$_L$-V$_L$ are the most, second, and least favored substrates, respectively, for the upper moiety. Unlike the upper moiety, the specificity for the lower pyrroloindoline moiety is more flexible and can accept in total eight tryptophan-containing cyclodipeptides, including cW$_L$-A$_L$, cW$_L$-P$_L$, cW$_L$-V$_L$, cW$_L$-I$_L$, cW$_L$-L$_L$, cW$_L$-M$_L$, cW$_L$-F$_L$, and cW$_L$-Y$_L$.

Besides the capability of taking various substrates, NascB also shows tolerance in the regio- and stereo-specificity of the connection manner and C$^2$–C$^3$ stereo-configuration. The tolerance of these specificities is increased in the order of cW$_L$-P$_L$ < cW$_L$-A$_L$ < cW$_L$-V$_L$ containing products: (i) In the cW$_L$-P$_L$ containing products, both of the regio- and stereo-specificities are conserved and the same as those observed in NAS-C. (ii) In cW$_L$-A$_L$-containing products, six products (**NAS-1**, **4**, and **6–9**) share the same the regio- and stereo-specificity to NAS-C. Only two products **NAS-2** and **NAS-5** show an identical specificity to the NAS-A/B (C$^3$–C$^{7'}$ bond, 2R-3S), and one product with a not previously observed combination of bond and stereo-configuration (**NAS-3**, C$^3$–C$^{7'}$, and 2S-3R). (iii) In cW$_L$-V$_L$-containing products, only four products (**NAS-18**, **20**, **22**, and **25**) retain the specificity of NAS-C and three (**NAS-17**, **23**, and **26**) have a specificity of NAS-A/B. Three remaining products show a combination of C$^3$–C$^{6'}$/2R-3S (**NAS-24**) and C$^3$–C$^{7'}$/2S-3R (**NAS-19** and **21**), which have not previously been discovered. This substrate tolerance enables a single enzyme transformation to produce analogs with different specificities, which is very efficient for generating structural varieties. Interestingly, in some

products, the sulfur group of their methionine moiety was oxidized to a sulfoxide (**NAS-7** and **14**) or sulfone (**NAS-8**) group. This spontaneous or enzymatic oxidation (by endogenous enzymes) further increases the structural varieties of the NAS scaffold.

Since the enzyme shows very good substrate flexibility, we were next interested in their reaction on halogenated and D-residue containing substrates. We chemically synthesized six halogenated substrates including 7F-, 5Cl-, 6Cl-, 7Cl-, 8Cl-, 6Br-cW$_L$-P$_L$, and three D-residue containing cyclodipeptides including cW$_D$-P$_D$, cW$_D$-P$_L$, and cW$_L$-P$_D$ (**s21**–**s29**, Supplementary Figure 5). Then, we individually fed each of the six halogenated and three D-amino acid containing cyclodipeptides to the recombinant *E. coli* GB05dir-T7-NascB. The enzyme only recognizes halogens in the 7-position, and both of 7F- and 7Cl-cW$_L$-P$_L$ can be efficiently converted into their corresponding products (**NAS-27** and **28**, Fig. 4, Supplementary Figure 6d, e, Supplementary Tables 1–4). Interestingly, these two halogenated products show a different stereo-configuration in C$^2$–C$^3$ from each other, although they have the same C$^3$–C$^{6'}$ bond. The D-residue containing substrates are not able to form homo-coupled products, while, through another co-fed experiment (Supplementary Tables 2–4), they can be efficiently coupled with cW$_L$-V$_L$ to form hetero-pyrroloindolines (**NAS-29** and **30**). This is surprising as natural C$^3$-aryl pyrroloindolines are very rare with D-amino acids and this result indicates the potential for expanding the structural variety through the incorporation of D-amino acids.

**Bioactivity assessment**. The bioactivity of NASs are rarely reported: (i) NAS-A/B have no cytotoxicity to arrays of bacteria, fungi, or cancer cell lines[4]. (ii) NAS-C shows only a trivial activity against the chloroquine-sensitive malaria parasite, *Plasmodium falciparum*[5]. Considering that many alkaloids are active on neuronal systems and NASs analogs are structurally similar to alkaloids, we are curious to know whether NASs analogs are also effective on neuronal systems. To evaluate this potential bioactivity, both the Aβ25-35-induced and L-glutamic acid-induced PC-12 cell lines were prepared and used as a model of Alzheimer and nerve injury, respectively. Compounds **NAS-1** to **-28** were subjected to testing in these two models. The results indicated that these products have no activity on the Alzheimer model (Supplementary Table 5), while most products exhibited an obvious protection activity against glutamate-induced PC-12 damage (Supplementary Table 6). Particularly, **NAS-12, 27, 10**, and **11** show a potent activity, which is even better than the control nimodipine (Table 1), a clinical drug used for nerve protection. Interestingly, these products all bear a proline in the upper moiety and C$^3$–C$^{6'}$ bond, suggesting both the upper moiety and regio-specificity are critical for bioactivity, while the groups in the lower pyrroloindoline moiety are less important.

**Table 1 The protective effects of some most potent NAS analogs against glutamate-induced PC-12 cell apoptosis**

| Compound | Concentration (μM) | Inhibition rate (%) |
|---|---|---|
| **NAS-12** | 10 | 29.01 ± 1.67 |
| **NAS-27** | 10 | 29.88 ± 2.87 |
| **NAS-10** | 10 | 32.73 ± 2.83 |
| **NAS-11** | 10 | 33.00 ± 4.92 |
| Nimodipine (positive control) | 20 | 33.30 ± 2.79 |
| Negative control | 0 | 42.04 ± 4.86 |

## Discussion

Biocatalysis is often more efficient and cost-effective than chemical routes. The selectivity of reactions is principally induced by the microenvironment in the catalytic cavity of an enzyme, so enzymatic conversions can often generate chemically unfavored stereo- and regio-selectivity in very high degree. These merits make biocatalysis a very appealing route for the generation of pyrroloindoline-containing compounds with unfavored stereo and regio-selectivity.

The above-presented results show that NascB is clearly able to generate all four combinations of two regio-specific and two stereo-specific reactions. Its unusual substrate-binding mechanism is interesting and significant for further engineering of specificity. NascB is assumed to contain two independent pockets for accommodation of the upper cyclodipeptides and lower pyrroloindolines moieties, respectively: the upper pocket accommodating the upper cyclodipeptide, while the lower pocket accommodates the lower pyrroloindoline. To generate both the C$^3$R and C$^3$S products, the heme-containing lower pocket is necessary to grant substrates freedom to expose the *Re* or *Si* face of C$^3$ to the other substrate molecule in the upper pocket. In addition, this pocket is highly tolerant to structural variations in the C$^{15}$-substituted groups of the pyrroloindolines moieties (Fig. 1), as bulky residues such as methionine and phenylalanine can be accepted. The failure to take polar residues (except for tyrosine) suggests that the chemical environment in this pocket is indeed very hydrophobic, which could be optimized by protein engineering to further broaden its substrate scope to include polar residues or other unnatural groups. In contrast to the lower pocket, the upper pocket is apparently more crowded and hydrophobic as only three small cyclodipeptides including cW$_L$-P$_L$, cW$_L$-A$_L$, and cW$_L$-V$_L$ can be accepted.

Analysis of the conformation of each product revealed that the upper cyclodipeptide units are all perpendicular to the pyrroloindoline moiety and the small residue (or not Trp residue) is pointing away from the pyrroloindoline (Supplementary Figure 10). Interestingly, both the scaffolds C$^3$–C$^{7'}$/2$S$-3$R$ (**NAS-5**) and C$^3$–C$^{6'}$/2$R$-3$S$ (**NAS-24**), particularly the latter, can cause severe steric interaction between the upper and lower small residues (Supplementary Figure 10), thus are thermodynamically unfavored. Chemical synthesis of the scaffolds with 2$S$-3$R$ is very difficult as these are normally dependent on the induction of a particular cyclodipeptides conformation[7], while this biosynthetic approach provides a practical and efficient way to access these. To allow NascB to accept more diverse structures in the upper pocket, further protein engineering may be employed to increase the available space. Moreover, reduction of the size of the upper moiety such as using smaller phenylalanine-containing cyclodipeptides or even indole groups are also promising avenues to expand pyrroloindolines diversity.

The biosynthesis of many natural products biosynthesis features C–C bond or C–N bond-coupling formation[13,14,17,24–31]. These reactions are often radical-mediated, processed by Flavin-dependent, SAM-dependent, metal-dependent enzymes, or P450. Due to high reactivity, radicals can accomplish very difficult chemical transformations. However, their instability, destructive effect, and short-life in normal conditions make it very challenging to tame their reactivity. Unlike other radical reactions, the NascB-catalyzed reaction presents a rare radical cascade reaction and involves a three-steps conversion: including the radical generation and migration, Mannich reaction, and radical addition. Such a multistep radical-involving transformation is very efficient and our whole-cell biocatalysis system makes NascB ideal for practical application.

The absence of an efficient production approach restricts the exploration of biological activities of pyrroloindoline-containing compounds with this characteristic molecular architecture. So far,

only trivial bioactivity has been known through testing a limited set of molecules. With our enzyme-based and efficient biocatalysis platform, it is now possible to significantly expand the bioactive space of heterodimeric $C^3$-aryl pyrroloindolines.

## Methods

**Heterologous expression of biosynthetic gene clusters**. We extracted the genomic DNA of the NAS-producing strain of *Streptomyces* sp. (CMB-MQ030) through a method developed by Nikodinovic et al.[32] with minor modification. This minor modification is one more round of chloroform treatment before isopropanol precipitation of genomic DNA. To sequence the genome of the NAS-producing strain, two SMRT cells were employed at UQ Centre for Clinical Genomics (UQCCG) to generate 114,142 reads with a mean read length of 14,421 bp, which provided an average of ×194.7 coverage across the genome reference. The finished genome was assembled with HGAP2[33]. The gene clusters were amplified by PCR (Supplementary Methods) and inserted into the vector pIB139 under the driving of the constitutive promoter (*ermE**) for heterologous expression in the model strain *Streptomyces albus* J1074.

**Protein expression, purification, and enzyme assay**. P450 genes with and without codon optimized for *E. coli* were cloned into pET28a and overexpressed in *E. coli* BL21 (DE3). NascB activity against $cW_L$-$P_L$ was assayed by incubating purified NascB (0.1 μM) and $cW_L$-$P_L$ (1 mM) at 18 °C with 1 μM *E. coli* flavodoxin (FdX), flavodoxin reductase (FdxR), or 1 μM spinach ferredoxin (Fd), ferredoxin reductase (FdR), 2 mM NADP$^+$ (Sigma-Aldrich), 2 mM glucose, and 2 mM glucose dehydrogenase (GDH) in 50 mM HEPES buffer, 100 mM NaCl, at pH 7.5. The reaction was left at 18 °C for 24 h. After 24 h, two times more volume of ethyl acetate was added into the reaction solution, followed by the sonication for 5 min. After the separation of aqueous and organic phase, the top ethyl acetate was transferred to a rotavapor to dry, which was re-dissolved in HPLC-graded methanol with an addition of small amount of DMSO, if the solubility is poor. The resultant solution was filtered through 0.45 μM membrane and subjected to analysis by HPLC analysis. A diamonsil (C18, 5 μm, 250 × 4.6 mm, Dikma Technologies Inc.) was used with a flow rate at 1 mL min$^{-1}$ and a PDA detector over a 40 min gradient program with water (eluent A) and acetonitrile (eluent B): $T = 0$ min, 5% B; $T = 30$ min, 100% B; $T = 33$ min, 100% B; $T = 34$ min, 5% B; $T = 40$ min, 5% B.

**Electron paramagnetic resonance spectroscopy**. X-band CW EPR spectra were recorded at 15 K on a Bruker Elexsys E500 spectrometer fitted with a super high Q Bruker cavity, a liquid helium cryostat (Oxford Instrument, and a microwave frequency counter (BrukerER049X). Spectra were measured with a microwave power of 2 mW (non-saturating conditions), a modulation amplitude of 0.3 mT, and a modulation frequency of 100 KHz. The magnetic field was calibrated with a Tesla meter. Frozen solutions of the substrate-free and substrate-bound enzyme in 50 mM Tris-HCl, pH 7.5, 100 mM NaCl, 10% glycerol were analyzed in 4 mm-quartz tubes. For the substrate test, 80 μL of 0.23 mM NascB solution was mixed with 0.368 μL of 100 mM $cW_L$-$P_L$ solution in DMSO to obtain a twofold substrate excess in the final mix. For the substrate-free test, the same amount of NascB enzyme solution was mixed with 0.368 μL of DMSO.

**Biotransformation and purifications**. The recombinant strains were prepared by growth to an OD$_{600}$ of 0.8–1.0 at 37 °C. After expression at 18 °C, 220 r.p.m. for 20 h under 100 μM IPTG (isopropyl-β-D-thiogalactopyranoside), 0.4 mM δ-aminolevulinic acid (ALA), and 0.2 mM $(NH_4)_2Fe(SO_4)_2$ induction (they were added when OD$_{600}$ reached 0.8–1.0), cells from 50 mL culture were harvested by centrifugation at 2000×*g* at 4 °C and were washed twice and resuspended in 2 mL M9 medium. Then cyclodipeptide substrates were added in the M9 medium. After 48 h incubation at 18 °C, the reaction mixture was extracted with 4 mL ethyl acetate under sonication for 10 min. Organic phase was transferred and dried by vacuum at low temperature. Metabolites were subsequently re-dissolved by methanol with the right amount of DMSO and filtrated by a 0.45 μm membrane to remove particles before HPLC or HPLC-MS (mass spectrometry) analysis.

**NMR spectroscopy**. The NMR spectra were recorded on a Bruker Avance III spectrometer at a $^1$H frequency of 400 MHz, 700 MHz, or 900 MHz equipped with a TCI cryoprobe. Lyophilized samples (varying from 1 to 7 mg) were dissolved in 280 μL DMSO-d6 (Cambridge Isotope) and all spectra were recorded at 25 °C (298 K). $^1$H and $^{13}$C resonances were assigned through the analysis of 1D–$^1$H, 1D–$^{13}$C, 2D $^1$H–$^1$H rotating frame Overhauser effect spectroscopy (ROESY), 2D $^1$H–$^{13}$C heteronuclear single quantum correlation (HSQC), and 2D $^1$H–$^{13}$C heteronuclear multiple bond correlation (HMBC) (optimized for long-range heteronuclear couplings of 6 Hz). $^1$H and $^{13}$C chemical shifts were calibrated with reference to the DMSO solvent signal (2.50 and 39.5 p.p.m. for $^1$H and $^{13}$C, respectively). NMR experiments were processed with Bruker Topspin program (version 3.57) and analyzed with mnova software.

## Data Availability

The sequence of the *nasc* reported in this paper has been deposited in GenBank under accession number MH201515. The hyperlink of MH201515 is https://www.ncbi.nlm.nih.gov/nuccore/mh201515, which is currently on hold and will be released upon publication. All other relevant data are available from the corresponding authors.

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

## Acknowledgements

We thank Prof. Rob Capon and Dr. Zeinab Khalil at The University of Queensland (UQ) for the assistance and transfer of *Streptomyces* sp. CMB-MQ030 strain and NAS-A and -B standards; Professor Youming Zhang at Shandong University for gifting the *E. coli* GB05dir. This work was supported in part by the NSFC (31570057 and 31770063 to X.Q.) and UQ (UQ Postdoctoral Research Fellowship 2015–2017 to X.J., Development Fellowship 2017–2019 to M.M.). We thank the Queensland NMR Network for access to the NMR spectrometer and Centre for Advanced Imaging for access to the EPR spectrometer.

## Author contributions

X.Q., X.J., and Z.D. designed research; W.T., C.S., X.J., M.Z., Y.Z., H.P., J.R.H., M.M., and D.Z. performed the experiment; M.Y. and S.-L.C. perform the DFT calculation; X.Q., W.T., and C.S. analyzed data; and X.Q., W.T., C.S., and X.J. wrote the paper.

## Additional information

**Competing interests:** The authors declare no competing interests.

