## [Peer Review File · Nature Communications]

Reviewers' comments:

Reviewer #1 (Remarks to the Author):

This is a very nice collaborative paper by the Qu, Jia and Deng work groups, which elucidates a synthetically challenging C-C bond formation catalyzed by a cytochrome P450 monooxygenase. Specifically, the authors studied the biosynthesis of a complex alkaloid framework that can be found in various biologically active natural products such as the naseseazines. They found that a novel CYP450 enzyme promotes the regio- and stereoselective intramolecular and intermolecular C-C bond formation and deduced a model for the course of the reaction. More importantly, the authors succeeded in reconstituting the biotransformation in *E. coli*, which was obviously not a trivial task. Using this whole-cell system they were able to generate 30 novel NAS analogues through biotransformation of synthetic precursors and found that some of the analogues prepared have neuroprotective properties. This is a great example of merging the power of enzyme catalysis with synthesis, since the de novo total synthesis of such compounds would have been out of question. What I also really like about this work is that the authors obtained multi-mg amounts of the new compounds and fully characterized the new compounds. Often, only mass spectra are shown. Here, the authors show nice spectra that document the identity and purity of the compounds. Overall, the quality of the study is very high, and I have only minor comments that mainly refer to language and formatting. Otherwise I think the paper is well suitable for publication in Nature Communications.

Please present this work in a broader context and cite other examples on related C-C couplings and N radicals.

Note that flavoproteins may mediate mechanistically related couplings, see for example Baunach et al *Angew Chem Int Ed.* 2013. doi: 10.1002/anie.201303733. Please cite such papers, too.

Abstract: What are "interesting biological activities"?

Discussion: "with this fascinating molecular architecture": please tone down

Various formatting errors need to be fixed, e.g. space before °C, subtitles capitalized or not, etc.

Reviewer #2 (Remarks to the Author):

The paper by Jia, Qu, et al describes an investigation of the biosynthesis of the naturally-occurring hexahydropyrroloindole (HPI) naseseazine C (NAS-C). The authors proceed to develop a whole-cell biocatalysis system and use it to prepare a variety of structural analogues of NAS-C. These structural analogues would be very difficult to make by means of chemical synthesis. Some of these analogues are shown to have neuroprotective properties. This result demonstrates that the biosynthetic approach to NAS-C analogues can have practical applications in drug discovery. I find it particularly remarkable that the mechanism of NAS-C formation involves an addition of a C3-radical HPI intermediate to C6 position of indole in the uncyclized precursor. The authors present a convincing mechanistic picture that is supported by experimental studies (for example, influence of electron-withdrawing substituents), even though additions of such relatively stabilized tertiary radicals to indoles are without precedent. A question that appears to be left out of the mechanistic discussion is the possibility of formation of the N-protonated N1-radical (second intermediate in the N1-H abstraction pathway) directly

from cWL-PL by electron transfer. Has the ET step been explored and ruled out? While there is nothing unusual in N1-hydrogen abstraction that is followed by protonation, it will be interesting to see how it compares with the alternative pathway that involves ET and protonation of the Fe-O- to compound II.

Additionally, please correct "mannish" to "Mannich" on page 6.

Overall, I believe that the manuscript provides a new fundamental perspective on the biosynthesis of HPI-containing natural products that are an important class of alkaloids. The work also has important practical implications, as evidenced by preparation of numerous structurally diverse analogues of NAS-C with neuroprotective properties. I believe the paper can be published in Nature Communications.

Reviewer #3 (Remarks to the Author):

The authors of the manuscript titled "Efficient Biosynthesis of Heterodimeric C3-Aryl Pyrroloindoline Alkaloids" investigate the biosynthesis of nasesezine C (NAS-C), a naturally occurring hexahydropyrroloindole. The biosynthetic pathway of NAS-C was found in *Streptomyces* sp. CMB-MQ030 and near this gene cluster was a P450 enzyme that was postulated to play a role in the biosynthesis of this compound. The used X-band CW EPR to demonstrate that NAS-C biosynthesis involves radical-mediated intramolecular cyclization by this P450. They then heterologously expressed the genes (*fd*, *fdr* and *nascB*) for NAS-C biosynthesis in *Escherichia coli*, producing NAS-C and confirming the biosynthetic pathway. Additionally, the substrate scope of P450- NAS-C was tested by supplementation of the heterologous host with 20 chemically-synthesized L-Trp containing diketopiperazines and a number of analogues were produced. Finally, a bioactive assessment was performed and it was determined that NAS-12, 27, 10 and 11 showed activity against glutamate induced PC-12 cells apoptosis that is comparable to the control, Nimodipine.

This manuscript does not meet the novelty threshold required to be published in Nature Communications nor will it be of broad interest to the community. Nasesezine C is a previously discovered and characterized compound and its biosynthetic pathway is similar to NAS-A/B. The new analogues of NAS-C were obtained by simply adding a number of L-Trp derivatives. This work thus did not develop new broadly applicable methodology, challenge existing understanding, or characterize a fundamentally new mechanism of natural product chemistry or biosynthesis. As such the work will not make a significant scientific impact on the biosynthesis or natural product community. It will be of interest to a limited group of researchers with interest 1) in the mechanism of p450 biochemistry and 2) tryptophan containing natural products. The biological activity obtained from this small library with limited chemical diversity showed mild potency and will not be of broad interest to the medicinal chemistry or chemical biology community.

The manuscript could be significantly more concise.

Reviewer #1:

1. Please present this work in a broader context and cite other examples on related C-C couplings and N radicals. Note that flavoproteins may mediate mechanistically related couplings, see for example Baunach et al *Angew Chem Int Ed.* 2013. doi: 10.1002/anie.201303733. Please cite such papers, too.

Thank you very much. We have added a paragraph on Page 12 to discuss the related C-C coupling and N radicals. Also, we have cited this paper as reference 29 and other papers as references 30-33 in our discussion.

2. Abstract: What are "interesting biological activities"?

To avoid misunderstanding, we have deleted the word of "interesting" on page 1.

3. Discussion: "with this fascinating molecular architecture": please tone down

We have changed "fascinating" into "characteristic" on pages 2 and 12.

4. Various formatting errors need to be fixed, e.g. space before °C, subtitles capitalized or not, etc.

We apologized for those formatting errors. We did our best to find and fix those errors.

Reviewer #2:

1. A question that appears to be left out of the mechanistic discussion is the possibility of formation of the N-protonated N1-radical (second intermediate in the N1-H abstraction pathway) directly from cWL-PL by electron transfer. Has the ET step been explored and ruled out? While there is nothing unusual in N1-hydrogen abstraction that is followed by protonation, it will be interesting to see how it compares with the alternative pathway that involves ET and protonation of the Fe-O⁻ to compound II. Additionally, please correct "mannish" to "Mannich" on page 6.

Thank you very much for raising such a good point. We apologize for not considering the possibility of electron transfer (ET) and protonation of the Fe-O⁻ to compound II. Here, we will rule out the possibility from the perspective of the axial ligation. These histidine-ligated enzymes, such as horseradish peroxidase (HRP), typically perform one-electron or electron transfer (ET) oxidations.

In their thiolate-ligation counterparts (P450), thiolate ligation is a very strong axial electron donation to the iron center and promotes the process of H-atom abstraction through two mechanisms: (1) the creation of basic ferryls significantly decrease one-electron reduction potential of ≥ 0.3 V of compound I; (2) the elevated pK_a of compound II of \geq five units greater increases the driving force for H-atom abstraction (ref 15 and 16: *Current Opinion in Chemical Biology* 2009, 13:84–88 and *science*, 2004, 304, 1653). Those two references and one sentence have been appended on page 6.

In addition, Professor Houk, Garg, Tang and coworkers (pls see ref 17 Lin et al., *J. Am. Chem. Soc.* 2016, **138**, 12, 4002-4005) used extensive calculation to confirm that

N1-hydrogen abstraction in the N1 indole of tryptamine by the p450 enzyme should be more possible.

We have corrected "mannish" to "Mannich" on page 5.

Reviewer #3:

This manuscript does not meet the novelty threshold required to be published in Nature Communications nor will it be of broad interest to the community. Naseseazine C is a previously discovered and characterized compound and its biosynthetic pathway is similar to NAS-A/B. The new analogues of NAS-C were obtained by simply adding a number of L-Trp derivatives. This work thus did not develop new broadly applicable methodology, challenge existing understanding, or characterize a fundamentally new mechanism of natural product chemistry or biosynthesis. As such the work will not make a significant scientific impact on the biosynthesis or natural product community. It will be of interest to a limited group of researchers with interest 1) in the mechanism of p450 biochemistry and 2) tryptophan containing natural products. The biological activity obtained from this small library with limited chemical diversity showed mild potency and will not be of broad interest to the medicinal chemistry or chemical biology community.

We are sorry that we might not well present the novelty of our work in the manuscript. In terms of your points, following is our replies:

1. You mentioned: "Naseseazine C is a previously discovered and characterized compound and its biosynthetic pathway is similar to NAS-A/B. The new analogues of NAS-C were obtained by simply adding a number of L-Trp derivatives. This work thus did not develop new broadly applicable methodology, challenge existing understanding, or characterize a fundamentally new mechanism of natural product chemistry or biosynthesis." As such the work will not make a significant scientific impact on the biosynthesis or natural product community. It will be of interest to a limited group of researchers with interest 1) in the mechanism of p450 biochemistry and 2) tryptophan containing natural products.

It's true that Naseseazine C is a previously discovered and characterized compound, but both NAS-C and NAS-A/B biosynthetic pathways are previously not known. Within this work, we cloned and characterized the biosynthetic gene cluster of NAS-C for the first time. We found a unique P450 enzyme responsible for this type of scaffold formation. The mechanism of this enzyme is entirely unprecedented which involves a three-steps cascade, including a radical generation & migration, intramolecular Mannich-type cyclization and an intermolecular radical addition.

From the manuscript, you can see it is indeed very challenging to develop the whole cell biocatalysis system. We have spent more than half of a year on evaluating the effects of vectors and screening strains, and finally found an *E. coli* K-strain GB05dir that can tolerate the toxicity of P450. Because P450 reaction requires multiple associate proteins and expensive cofactors, a whole cell biocatalysis is the most ideal approach to implicate these reactions for a practical purpose. However, many p450 system shows cell toxicity, which forms a bottleneck for their practical implication. Our results not only solve the toxicity problem of NascB but also provides a potential universal solution to many other toxic P450.

Besides the catalytic mechanism, the substrate specificity of NascB is also very unusual. This enzyme can accept highly diverse substrates to generate large arrays of C3-aryl pyrroloindolines. The scaffold types of these products can be the combination of different region- and stereo- specificity. Besides the C³-C⁶ bond and with 2S-3R (NAS-C type), some products show a C³-C⁷ 2R-3S (NAS-A/B type), C³-

C^7 2*S*-3*R*, and C^3 - C^6 2*R*-3*S*. The latter two types are even not existing in nature before, which are also very challenging for chemical synthesis. Thereby, this enzyme is indeed highly useful to generate multiple types of C3-aryl pyrroloindolines. Considering these above mentioned merits, we believe our work both characterized a fundamentally new mechanism of natural product chemistry or biosynthesis, and developed a new broadly applicable methodology.

2. You mentioned: "The biological activity obtained from this small library with limited chemical diversity showed mild potency and will not be of broad interest to the medicinal chemistry or chemical biology community."

The number of the natural occurring heterodimeric C3-Aryl pyrroloindoline alkaloids is only five. Our study significantly increased this number to 30 which is six-fold larger than before.

More significantly, the neuron protection activity of these family is previously not known. Our study discovered this activity for the first time. The potency of some best novel analogues is indeed very strong. As shown in the Table 1, only half amount of NAS-12, 27, 10 and 11 can show a similar or better neuro protection activity than Nimodipine, the current best and clinic used drug for neuron protection!!

By assaying this library, we were able to obtain SAR relationship of NAS analogues which will extremely useful for further improve their activity.

Overall, we do believe our work represents a significant and systematic breakthrough in studying this important family of natural products. Our results will be of broad interest to the readership of research communities in the fields of biocatalysis systems, synthetic biology, medicinal chemistry and beyond.

REVIEWERS' COMMENTS:

Reviewer #2 (Remarks to the Author):

I believe the authors have responded to all of the questions raised by Reviewers and have made the required corrections. I think the manuscript can now be accepted without further changes.

Reviewer #3 (Remarks to the Author):

The manuscript entitled "Efficient Biosynthesis of Heterodimeric C3-Aryl Pyrroloindoline Alkaloids" by Tian et al describes the characterization of a P450 from *Nasoseazine* biosynthesis. The authors show that this can be used to couple a limited diversity of tryptophan containing diketopiperazines in a whole cell biocatalyst system. The isolated *Nasoseazine* analogs show modest activity as protection against glutamate-induced PC-12 damage.

The work described in this manuscript is of excellent quality and represents a significant effort on the part of the authors. The scientific conclusions are supported by the data. The work however does not meet the bar for significance to the readership of *Nature Communications*. It lacks novelty on the P450 mechanism, the chemical diversity accessible, and the biological activity of the new compounds.

This work is well suited for a more discipline specific journal where its technical excellence will be appreciated and it will be seen by the appropriate readership.

Concerns regarding significance.

Mechanism: P450 oxidation of indole containing diketopiperazine to generate pyrroloindoline is well preceded as is the further dimerization or herterodimerization via a radical mechanism (see Watanabe's work). In fact the authors' "entirely unprecedented ... three-steps cascade, including a radical generation & migration, intramolecular Mannich-type cyclization and an intermolecular radical addition" has been previously biochemically characterized in ditryptophenaline biosynthesis (*ChemBioChem* 2014, 15, 656). The conceptual difference in this study is the site of intermolecular radical addition, which in the current study occurs at C6 and C7 of indole. Oxidation of indole C6 or C7 by a p450 is well preceded (see Kim or Guengerich's work), thus the formation of the 6' and 7' adducts is supported by literature precedent. There are multiple examples of p450s leading to non-regioselective oxidation of indoles at adjacent sites, thus the product distribution between 6' and 7' adducts is also not unprecedented. It is a stretch to identify this biochemistry as entirely unprecedented. This work does not expand our understanding of the types of chemistry possible through P450 mediated oxidation, nor does it introduce an unusual or unexpected site of oxidation for indoles or indole containing diketopiperazines.

Chemical diversity: The chemical diversity accessed by the *in vivo* biocatalytic system is very limited even though a number of new compounds are made. The modifications are highly conservative with non-polar protienogenic amino acids being replaced by other non-polar protienogenic amino acids. The substrate tolerance is consistent with what is known about substrate binding in P450s. They typically have a non-polar pockets for substrate binding. Thus tolerating modified non-polar substituents is thus not unprecedented or unexpected for a P450.

Biological activity: None of the parent natural products show significant biological activity. Some of the analogs generated in this study show protection against glutamate-induced PC-12 damage. While this activity exceeds the observed activity for nimodipine, a clinical agent, MgSO₄ has also been shown to exceed the activity of nimodipine, reducing the significance of the reported NAS analog data.

Response to Reviewer #3:

We appreciate your concern with the significance to the readership of Nature Communications. However, we still disagree with your opinions on the novelty and significance of our work. Here is our point to point explanation.

You mentioned: *Mechanism: P450 oxidation of indole containing doketopiperazine to generate pyrroloindoline is well precedented as is the further dimerization or heterodimerization via a radical mechanism (see Watanabe's work). In fact the authors' "entirely unprecedented ... three-steps cascade, including a radical generation & migration, intramolecular Mannich-type cyclization and an intermolecular radical addition" has been previously biochemically characterized in ditryptophenaline biosynthesis (ChemBioChem 2014, 15, 656). The conceptual difference in this study is the site of intermolecular radical addition, which in the current study occurs at C6 and C7 of indole. Oxidation of indole C6 or C7 by a p450 is well precedented (see Kim or Guengerich's work), thus the formation of the 6' and 7' adducts is supported by literature precedent. There are multiple examples of p450s leading to non-regioselective oxidation of indoles at adjacent sites, thus the product distribution between 6' and 7' adducts is also not unprecedented. It is a stretch to identify this biochemistry as entirely unprecedented. This work does not expand our understanding of the types of chemistry possible through P450 mediated oxidation, nor does it introduce an unusual or unexpected site of oxidation for indoles or indole containing diketopiperazines.*

Response : The mechanism of ditryptophenaline biosynthesis (Prof. Watanabe's work on ChemBioChem 2014, 15, 656) is very different from ours. Their enzyme abstracts the H radical at C10 and then attacked the C2 of indole to form the C3-pyrroloindoline radical. Two molecules of C3-pyrroloindoline radicals then coupled each other to form the ditryptophenaline. Our p450 reaction involves an intramolecular Mannich-type reaction and intermolecular radical addition, both of which are not involved in ditryptophenaline biosynthesis.

It's true that oxidation of indole C6 or C7 by a p450 is precedented, however these reactions are hydroxylation reactions. For instance, in Prof. Guengerich's work (Biochemistry 2000, 39, 13817-13824), they demonstrated that their p450 can hydroxylate the C6 and C7 position of indole to produce the analogues of indigo. Prof. Kim's work is similar. Despite both reactions are oxidation, however hydroxylation reaction is obviously different from our radical (C3-pyrroloindoline) addition reaction. Nowadays, discovering enzymes with entirely new chemistry is almost impossible. The novelty of enzymatic reactions is mostly referred to i) novel specificity, e.g. p450 imposing on a new substrate or showing new region-/ stereo-specificity (Nat Chem. 2011, 3, 738-743), or ii) new reactivity, such as p450 initiates a sequential/cascade hydroxylation reaction (Nature Chem, 2011, 3, 628-633). Our p450 (NascB) has both new specificity and reactivity. Based on these reasons, we do believe the mechanism of our enzyme is bright new which is able to expand our understanding of the types of chemistry possible through P450 mediated oxidation.

You mentioned: *Chemical diversity: The chemical diversity accessed by the in vivo biocatalytic system is very limited even though a number of new compounds are made. The modifications are highly conservative with non-polar protienogenic amino acids being replaced by other non-polar protienogenic amino acids. The substrate tolerance is consistent with what is known about substrate binding in P450s. They typically have a non-polar pockets for substrate binding. Thus tolerating modified non-polar substituents is thus not unprecedented or unexpected for a P450.*

Response : For a selected enzyme reaction, stereo-specificity, tolerance of the size of substituent groups and polarity are three major aspects which are concerned for

their widely utility. In our work, we demonstrated our enzyme' flexibility in stereo-specificity and tolerance of the size of substituent groups. With this flexibility, we generated 30 novel analogues, which is SIX times as many as currently know natural occurring heterodimeric C3-aryl pyrroloindolines. These compounds are highly structurally complicated, which are very challenging for chemical synthesis. The generated large number of analogues and amenable promiscuous specificity of biocatalysis system do represent a significant advance for studying this compound family.

The flexibility of our enzyme in stereo-specificity and tolerance of bulky substituent groups are indeed good enough. As you mentioned, p450 enzymes typically have non-polar pockets for substrate binding, so protein engineering is often required to change their preference. Indeed, we have raised this point as an awaiting-to-solve issue in our manuscript. Current work provides a solid basis for further engineering the pocket to accept polar substrates.

You mentioned: *Biological activity: None of the parent natural products show significant biological activity. Some of the analogs generated in this study show protection against glutamate-induced PC-12 damage. While this activity exceeds the observed activity for nimodipine, a clinical agent, MgSO4 has also been shown to exceed the activity of nimodipine, reducing the significance of the reported NAS analog data.*

Response: Discovering a few new compounds better than a clinical used drug nimodipine is indeed an important deal. Moreover, comparing the neuron protection of different analogues, we achieved a Structure–Activity Relationship (SAR) which is significant for further improving their activity.

It is common that inorganic compounds are found to have the therapeutic applications. For example, Arsenic trioxide (As_2O_3), has been already used to treat leukemia in clinical. It is also understandable that $MgSO_4$ has good protection activity. Actually, one of the most important and challenging work for natural products study is to identify their bioactivity. Before this work, it is indeed completely not known for such family bearing neuron protection activity. Upon these reasons, we do believe our work is significant.